# Finance and Jobs: How Financial Markets and Prudential Regulation Shape Unemployment Dynamics

**Ekkehard Ernst**

Macro-economic policies and jobs unit, Research Department, International Labour Organization (ILO), 1211 Geneva, Switzerland; ernste@ilo.org

**Abstract:** This article explores the impact of financial market regulation on jobs. It argues that understanding the impact of finance on labor markets is key to an understanding of the trade-off between economic stability and financial sector growth. The article combines information on labor market flows with indicators of financial market development and reforms to assess the implications of financial markets on employment dynamics directly, using information from the International Labour Organization (ILO) datatabse on unemployment flows. On the basis of a matching model of the labor market, it analyses the economic, institutional, and policy determinants of unemployment in- and out-flows. Against a set of basic controls, we present evidence regarding the relationship between financial sector development and reforms and their impact on unemployment dynamics. Using scenario analysis, the article demonstrates the importance of broad financial sector re-regulation to stabilize unemployment inflows and to promote faster employment growth. In particular, we find that encompassing financial sector regulation, had it been in place prior to the global financial crisis in 2008, would have helped a faster recovery in jobs.

**Keywords:** unemployment in- and out-flows; financial market development; financial market reforms; reform scenarios

**JEL Classification:** J64; J23

---

## 1. Introduction

The global financial and economic crisis (GFC) that erupted in 2008 profoundly changed our view of how financial markets affect economic performance and stability. The bankruptcy of Lehman Brothers—one of the oldest investment banks in the United States—demonstrated the immense risks that the global financial system had accumulated over the previous decade and the disastrous consequences that a sudden shift in risk perception was able to produce for growth and jobs. In a matter of a few months, the global economic system moved from a state of a symbiotic relationship between finance and growth to one where financial market risks were seen as inimical to stable and sustained economic performance. Most observers at the time had simply overlooked the enormous risks that a fast-growing financial sector had built up, which gradually undermined the stability of the entire system. This article argues that a focus on the financial market impact on employment dynamics would have helped a better understanding of the link between financial sector development and economic stability. Looking carefully at how financial market development and regulation affect job creation and destruction, we demonstrate how such insights allow us to inform on-going discussions on financial sector reforms. Specifically, the article shows how financial sector regulation could have helped overcome the GFC-induced jobs crisis more rapidly had some of the currently-implemented regulations been in place already earlier.

Regulatory activity on financial markets has been intensive since 2009. Indeed, soon after the outbreak of the crisis, a debate arose regarding the most appropriate changes to the regulatory framework of financial markets. After an intense period of policy and regulatory action (see International Institute for Labour Studies (2010, Chp. 5) for a summary of regulatory measures) a new debate arose which questioned the implemented regulation. Regulatory reforms ratified by the U.S. administration in the Dodd–Frank act in 2010 were among the most encompassing that the financial industry had seen in recent history. In Europe, where policy makers also faced a sovereign debt crisis three years later, similarly bold changes are still in the process of being fully implemented. At the international level, new regulatory agencies, such as the European Stability Mechanism, have been entering the scene and are beginning to influence global capital markets. Some reforms such as the Basel III agreement on stricter capital requirements are expected to be fully implemented only by 2022 (see the discussion in Basel Committee on Banking Supervision, 2017, Basel III: Finalising post-crisis reforms, Basel, available at: https://www.bis.org/bcbs/publ/d424.pdf, for more information on the implementation process). Nevertheless, disagreement continues as to whether the scope of regulatory reforms has been sufficient or if—on the contrary—policy makers have gone too far and are preventing economic activity from expanding more forcefully.

This disagreement is a reflection of the dramatic shift in perceptions that followed after several decades of academic research, highlighting the benefits of well-developed financial markets for economic growth (see, for instance, Levine 1997, 2005; Rajan and Zingales 1998). The earlier literature not only stressed the importance of well-developed financial markets for the real economy, it also provided empirical evidence of the different transmission channels, favoring market-based finance over bank credit, supposedly less favorable for innovation and entrepreneurship (Acemoglu et al. 2006; Levine and Zervos 1998). Accordingly, policy makers implemented significant reforms with the view toward developing sophisticated financial markets. The outbreak of the financial crisis, however, made it obvious that the previous growth spurts that followed financial market reforms were often accompanied by increases in inequality, making the acceleration in growth imbalanced and eventually unsustainable. In particular, the period of increasing financial market deregulation saw a significant fall in the labor income share across OECD countries (see Figure 1). Observers began to realize that an important transmission mechanism linking financial markets to the real economy ought to be taken into account in order to foster a full understanding of the links between financial sector development and risk (Rajan 2010).

The latter points to a more fundamental short-coming in the discussion on the benefits of financial market development and the necessity of financial sector reforms: none of the existing research has looked specifically at the impact of financial markets on employment growth and job creation, or the particular impact financial market development might have on employment and earnings volatility. Implicitly, it is assumed that an understanding of the growth channel of financial market reforms is sufficient to grasp the full impact of the latter on labor markets and jobs. Partly, this has to do with the fact that to date, little is understood about the interactions between financial market development and job creation, with only a few theoretical papers addressing the issue (see, for instance, Ernst and Semmler 2010; Pagano and Pica 2012; Wasmer and Weil 2004, which look at the interaction of financial development and unemployment from different angles). Drawing on existing theoretical insights, this article addresses the issue empirically, looking into the effects of both financial development and financial market regulation on labor market dynamics. Specifically, using both empirical estimates and scenario techniques, we ask whether and how the implementation of certain post-2010 reforms prior to the crisis would have affected recovery in employment.

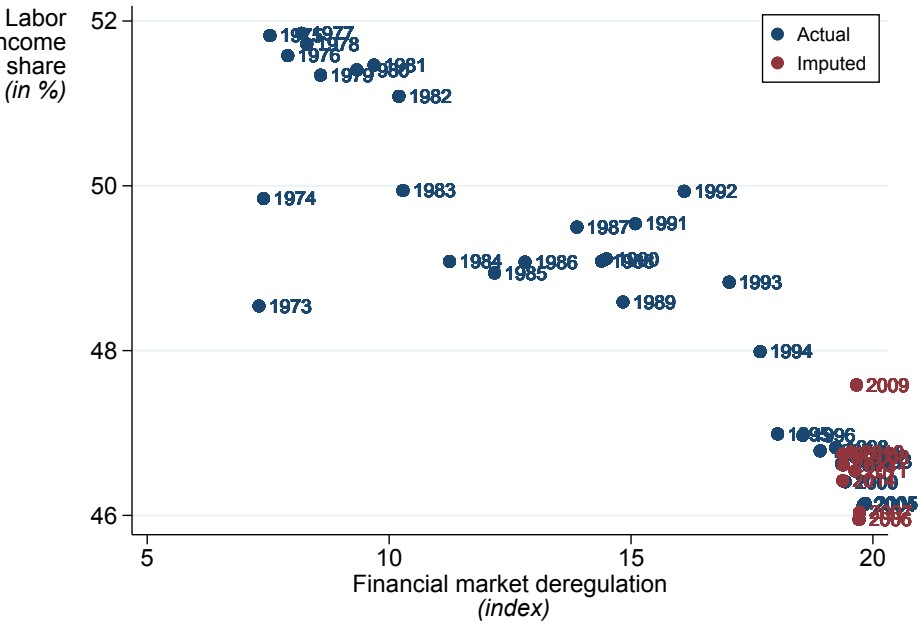

**Figure 1.** Labor income share and financial market reforms in the OECD (1973–2014). Note: The chart displays the average (employment weighted) labor income share in a sample of 20 OECD countries against the IMF financial market reform index between 1973 and 2014. The labor income share measures the some of wages as a share of nominal GDP (in percent). Values of the financial market reform index for years after 2005 have been imputed, using the method described in Section 5. Source: OECD Economic Outlook database, Abiad et al. (2008).

To address this question, the article analyses the impact of different aspects of financial market development and financial sector reforms by examining the gross margins of labor market adjustment prior to the global financial crisis. This allows us to understand the impact of financial markets on the dynamics of job creation and job destruction separately, using panel estimates for unemployment in- and out-flows based on a methodology originally developed by Shimer (2012) and later refined by Elsby et al. (2013) (see the discussion in ILO (2013b)). The article first presents evidence on the different margins of adjustment and their interaction with financial market characteristics. Combining different information sources regarding the evolution of financial sector regulation, this article also confirms the disruptive nature of financial market deregulation for the post-2009 period. Finally, carrying out a scenario analysis, the article then examines how different forms of financial sector reforms would have impacted employment dynamics, had they already been in place by 2010. The main results of the analysis can be summarized as follows:

- Financial market developments have a significant albeit ambiguous influence on unemployment dynamics. In particular, market-based financial development (both stock and bond markets) appears conducive to more labor market turbulence with higher unemployment in- and out-flows. On the other hand, greater international financial openness has the opposite effect, again with an ambiguous effect on overall unemployment rates.

- Regarding financial market reforms, securities market liberalization also leads to higher labor market turbulence, confirming the effect of the de facto development of stock and bond markets. Similarly, banking sector reforms such as loosening credit controls and banking sector privatization strengthen job creation without affecting unemployment inflows. At the same time, improved prudential regulation of banks leads unambiguously to lower unemployment as both unemployment outflows increase and unemployment inflows decline. In contrast to de facto

international openness, de jure capital account openness has an unambiguous positive effect on employment by increasing unemployment outflows and lowering unemployment inflows.

- Looking at the post-crisis period, financial market re-regulation appears modest when assessed in a historical context. Only a few additional constraints have been introduced that make financial regulation appear less market-friendly. When assessing the evolution of financial sector deregulation beyond 2009, financial market reform continues to display a distinctive pattern of increased labor market turbulence. Only the removal of entry barriers to the entry of new banks seem to reduce labor market turnover, albeit at the cost of an overall reduction in employment dynamics. Taken as a whole, financial market re-regulation had only a very limited positive effect on labor market developments.

- A scenario analysis suggests that much higher benefits for jobs could have been obtained had financial market reforms been bundled into a substantive regulation package. Adopting a political economy perspective, we present reasons why such an encompassing reform package has not been adopted despite its obvious labor market benefits. We demonstrate the potential positive impact on employment that a fully-fledged financial market reform could have had in comparison to a no-reform benchmark, but also with respect to partial reforms such as those effectively observed.

This article is structured as follows: The next section provides an overview of the existing literature on the effect of financial market development on labor markets. Section 3 introduces a simple model of labor market flows on the basis of a standard search and matching framework, extended by a more general description of shocks and taking into account capital accumulation at the firm level. Section 4 discusses the data and methodology employed in this paper. Section 5 presents the main results and examines the relative contribution of individual factors to the overall variation in our sample; it also presents evidence of the impact of financial market deregulation on labor markets after 2009 against an extended set of financial reform indicators using imputation techniques. Section 6 discusses some scenarios for financial market reform and their likely (combined) effect on labor market outcomes. A final section concludes.

## 2. Finance and Regulation: An Overview of the Literature

Prior to the global financial crisis, the rich literature on financial market development and growth focused mainly on the impact of the different characteristics of financial markets, sectoral specialization, and the institutional environment. The focus shifted significantly with the crisis, with an emerging literature paying increasing attention to the impact of financial markets on economic stability and the regulatory challenges financial market development pose. This section reviews some of the earlier literature on finance and growth, discusses its more critical reception since the crisis, and discusses some considerations that focused specifically on labor market issues. It also presents a selection of the vast and rapidly-growing literature on financial market regulation.

### 2.1. Finance and Growth

A substantial body of research has pointed to the benefits of financial development for growth, arguing that the financial sector plays an important role in mobilizing savings (Levine 1997; Levine and Zervos 1998), as well as in providing essential risk-sharing services (e.g., Allen and Gale 1999). Identification is not straightforward, however, as standard financial market variables such as credit growth or stock market valuations co-move with economic growth, which prevents the use of standard OLS-type estimations. To address this issue, different identification methods have been developed and applied (Beck 2009). The most common approach has been to use panel instrumental variables, often making use of sectoral data to identify sectors with a particular dependence on external finance. Some authors have also applied country-specific time-series approaches, exploring the forecast capacity of financial development for future growth rates (e.g., Rousseau and Wachtel 1998). Finally, some authors have made use of firm-level data to gain a better understanding of the particular

transmission mechanisms through which financial development can affect economic performance (e.g., Love 2003). Levine (2005) and Demirgüç-Kunt and Levine (2001) provide good overviews of both the theoretical and empirical literature that developed in this area prior to the GFC.

Looking at (instrumented) cross-sectional data, Beck et al. (2000) identified particular channels through which financial development affects growth. In particular, they found that financial intermediaries raise growth by lifting total factor productivity growth, whereas these institutions only weakly impact either physical capital growth or private savings rates. In this regard, Aghion et al. (2005) argued that financial development does not only help accelerate growth, it is also particularly beneficial for developing countries to catch up with more advanced economies; the further away a country is from the production frontier, the more financial development will help in closing the gap. Financial institutions seem to play different roles, however, in the catch-up process, as argued by Acemoglu et al. (2006). Whereas financial intermediaries are particularly important in providing capital to companies that are in the process of adopting technologies from the world frontier, market-based finance becomes important in selecting innovative firms to help push out the technological frontier. Therefore, different financial market institutions are supporting countries as they progress from emerging to developed economic status. In contrast, Compton and Giedeman (2011) took the view that financial intermediaries, such as banks, are complementary to growth in countries with weak governance structures (such as high levels of corruption and an absence of the rule of law), but do support growth in countries with higher institutional quality. In their view, only stock markets have a significant and independent impact on growth regardless of the quality of institutional development.

Alternative approaches have aimed at identifying the transmission channels of financial development on the real economy through sectoral analysis. An approach put forward by Rajan and Zingales (1998), and later extended by Carlin and Mayer (2003), commenced from the premise that sectors rely differently on external financing. The specific sectoral panel data setup allowed the authors to identify a growth effect from more developed financial markets for those sectors that rely more heavily on external financing. Specifically, Carlin and Mayer (2003) confirmed the results from aggregate estimates, showing that the growth effect of financial development comes from higher expenditures in R&D rather than from fixed capital formation. In contrast, Rousseau and Vuthipadadorn (2005) argued that for Asian economies, the investment channel has been the more relevant transmission mechanism in their catching up phase over the 1950–2000 period.

Since the GFC, research has focused on the possibility that there might be an upper limit to financial sector growth, above which negative effects of finance on economic stability prevail. Whereas earlier research suggested that the growth-enhancing effect of financial liberalization outweighs its costs in the form of a higher risk of crises (Rancière et al. 2006), Arcand et al. (2015) argued that there is a threshold above which financial depth no longer has a positive effect on economic growth. In particular, the authors estimated that financial depth depresses output growth when the credit ratio exceeds one year of GDP. This threshold not only operates for advanced economies, but can be detected even for emerging economies in Asia and Latin America: Aizenman et al. (2015) presented evidence suggesting that there might be a tipping point above which further increases in financial development lead to reductions in growth, in particular in specific sectors. Loayza et al. (2018) provided an overview of the literature looking specifically at the trade-off between faster growth and higher crisis risk. Even though the growth effects outweigh the crisis risk for middle-income countries, the challenges of "too much finance" are present for a large group of countries, in which overly-large financial sectors crowd out productive activities and lead to misallocation of resources. Echoing this pessimistic view, Intartaglia et al. (2018) presented evidence that all different forms of debt—private household and non-financial corporation debt—appear to be harmful to economic growth, albeit to different degrees and depending on whether the country is advanced or developing. In particular, the total amount of debt seems to have a negative impact on growth only in developed economies, but not in developing countries, according to their estimates.

So far, no agreement seems to have been reached as to the particular reasons for the existence of such a threshold. Blanchard et al. (2016) focused on the role that international financial integration plays in destabilizing economies. In particular in emerging economies, large capital inflows seem to be concomitant with financial sector development that leads to pro-cyclical effects of credit growth, thereby both supporting economic growth and raising the risk for sudden stops and crisis. Such effects had already been detected in earlier periods during the integration of less well-developed European countries into the single market (Brezigar-Masten et al. 2008). Others have focused on the impact that financial market liberalization can have on inequality (Ernst and Escudero 2008; Kumhof et al. 2015; Rajan 2010): As gains from financial market development are distributed unequally, misallocation of credit can endogenously produce higher instability and increase the risk of sudden stops. It is, in particular, this latter mechanism that points to a particular role for labor market dynamics in understanding the trade-off between between economic growth and crisis that financial sector development is likely to influence.

*2.2. Financial Market Development and Employment Dynamics*

In line with the literature on the impact of finance and growth, several studies have looked more directly at the impact financial sector turbulences have had on employment creation. In particular, in the aftermath of the GFC, researchers aimed at a better understanding of the role banking sector problems play in explaining the fall in employment. Using firm-level employment data for Spain, Bentolila et al. (2017) analyzed the credit supply shock as banks were forced to restructure, indicating that around 24% of job losses were due to firms being attached to weak banks. Berton et al. (2018) showed similar evidence for Italy, highlighting the heterogeneous employment effect of financial shocks, with effects of financial shocks depending on the education, age, gender, nationality, and type of contract of job holders. This is in line with earlier evidence presented by Caggese and Cunat (2008), demonstrating that in Italy, financially-constrained firms make more use of temporary employment than unconstrained ones, amplifying the employment volatility of shocks. The volatility-enhancing impact of banking crises was also confirmed in a panel of OECD countries prior to the crisis (Pagano and Pica 2012). In particular, during banking-related crises, job creation is more tightly linked to falls in output, thereby amplifying the employment impact of the recession (see Figure 2).

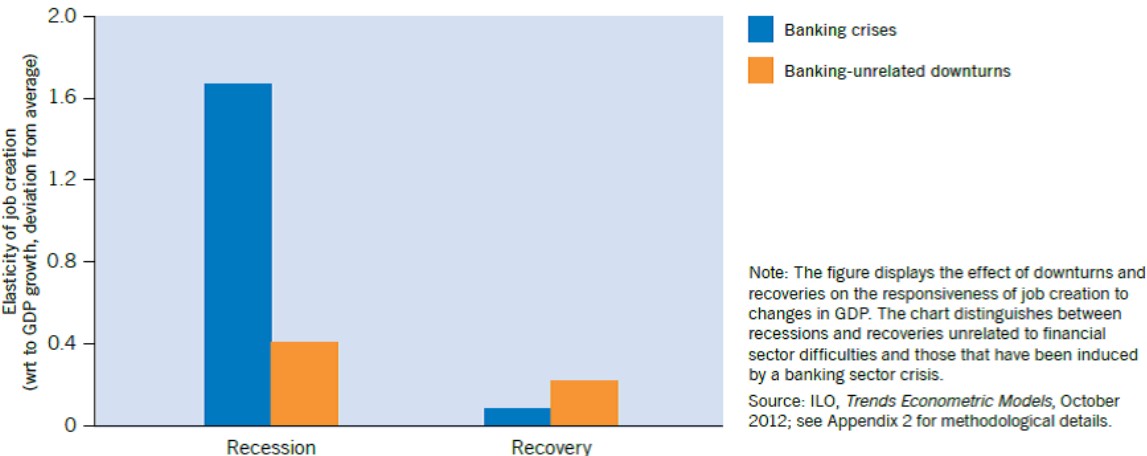

**Figure 2.** Banking crises and job creation. Source: ILO (2013a), p. 52.

Another strand of the literature has stressed the role of the financial premium and the financial accelerator in explaining labor market outcomes. Following considerations by Phelps (1998), for instance, Zoega (2012) stressed the role of asset prices in understanding employment dynamics. Looking specifically at the cyclicality of the external finance premium, Chugh (2013) demonstrated that credit-market frictions amplify aggregate total factor productivity shocks, leading to large employment

fluctuations that might help macro-economic models to generate realistic volatility in vacancies and job creation.[1] Part of the observed amplification of shocks has to do with the fact that larger banking sectors lead to more pronounced swings in the external finance premium (Epstein and Shapiro 2019). In addition, Gatti et al. (2012) showed that the impact of financial variables also depends on the specific rigidities present in the labor market: increased stock market capitalization enhances employment volatility in particular when labor markets are flexible. Finally, asset price effects might also operate in housing markets, amplifying shocks through a misallocation of resources between the housing market and the rest of the economy, thereby making adverse shocks to unemployment more persistent (Ernst and Saliba 2018).

In this regard, Wasmer and Weil (2004) have developed a general framework for better understanding the particular transmission mechanism that links financial shocks to labor markets. Introducing matching frictions also on financial markets, in their model, entrepreneurs need to search for both financial funds and workers. The complementary nature between capital and labor in this model causes financial frictions to magnify those on the labor market, thereby exercising additional upward pressure on equilibrium unemployment rates. Moreover, changes in capital costs, the availability of funds, or the nature of the (financial) matching process will have immediate repercussions on labor market outcomes, both in terms of employment and wages. On the basis of this theoretical framework, a small macro-economic literature has developed that tries to understand the role of the financial sector on the transmission of growth on employment. For instance, Ernst and Semmler (2010) showed that when the cost of issuing bonds changes endogenously due to wealth effects, shocks on output and employment are magnified, possibly leading to multiple equilibria and catastrophic job losses. In an empirical application of this approach that makes use of multi-regime vector auto-regressions, Ernst et al. (2016) demonstrated that shocks to credit conditions during a high-growth period have markedly different effects than during recessionary periods, amplifying the initial shock on output and employment.

*2.3. Financial Market Regulation: Between Economic Development and Instability*

Banking sector stability, therefore, appears to be key to ensuring that financial market development will play a positive role in the real economy. As highlighted by Monnin and Jokipii (2013) in a panel VAR for a sample of 18 OECD countries, a positive link exists between the stability of the banking sector and real output growth. When the banking sector is unstable, uncertainty about future output growth increases, depressing investment and employment. In addition, wealth effects resulting from the financial accelerator need to be specifically targeted through macro-prudential regulation. Following the debate on financial sector regulation after 2009, Claessens and Kodres (2014) summarized a few key lessons to restoring financial sector stability, including the adoption of a systemic approach, the introduction of crisis management as a an integral part of macro-economic policy making and taking into account regulatory arbitrage.

The latter points to political economy considerations that will also play a role in our scenario analysis. Indeed, as Pagano and Volpin (2005a) and Pagano and Volpin (2005b) have pointed out, in economies with well-developed financial markets, managers and financial investors have an incentive to lobby for lax financial market regulation together with flexible labor markets that allow keeping wages low. In particular, such a mechanism would explain the negative correlation observed in Figure 1. It is, therefore, no surprise that representatives from financial institutions have argued against overly-strict sector regulation, highlighting potential negative implications for activity levels (Institute of International Finance 2010). In contrast, academics and representatives from civil

---

1    Standard macro-economic models that abstract from financial frictions often have difficulties in replicating observed employment volatility, which has been dubbed the "Shimer puzzle", following Shimer (2005). Alternative specifications of the labor market matching process exist that can accommodate this volatility without resortingto counter-cyclical external finance premiums.

society have pointed out that previous increases in capital requirements have had little impact on credit growth, but helped make the economy more stable (Admati et al. 2013; Deli and Hasan 2018; Kashyap et al. 2010). Most of this work has concentrated on the impact of financial reforms on GDP growth and output levels, but so far, the labor market implications have rarely been part of a more detailed analysis, implicitly assuming a more or less fixed relationship between output and employment.

Recently, some research has emerged that tries to analyze the impact of macro-prudential regulation on labor markets more directly, demonstrating the role of self-employment as a particular transmission mechanism of volatility that prudential regulators need to target (Shapiro and Gómez 2015, 2017). So far, however, labor market specificities have played only a small role in understanding the effectiveness of financial market regulation on promoting growth and stabilizing the economy. This seems to be a major short-coming of the existing literature, given the prominent role that labor markets can play in amplifying shocks. To address this gap in the literature, this article offers a more direct analysis of the effects of financial sector regulation on unemployment dynamics. Starting from the seminal contribution by Wasmer and Weil (2004), we develop an empirical application of their framework in order to understand three essential dimensions of financial sector regulation as discussed in the literature reviewed thus far:

- Regulation of international capital flows and the international financial architecture will affect the availability of foreign capital in the domestic economy, lifting competition on the domestic banking sector and providing additional liquidity. This will help reduce domestic financial market frictions and should in principle lower the real long-term interest rate. At the same time, at the macroeconomic level, it can also increase economic volatility, in particular when financial deregulation is met with poor macroeconomic management and exchange rate misalignment. Over the long-run, this needs to be matched against the potentially positive effects of better integrated financial markets that can help especially low-income countries to overcome capital shortages and thereby promote job creation and limit job destruction.

- Banking sector regulation is expected to lead to lower credit growth and potentially to higher real interest rates. The former will reduce the growth rate of aggregate demand, dampening gross fixed capital formation and growth in disposable household income. In addition, the interest rate effect will not only slow down aggregate demand growth, but will directly affect the value of a new job, thereby lowering the rate at which new vacancies are being created and increasing the job destruction rate. At the same time, prudential banking sector reforms are expected to stabilize the financial sector, thereby raising the prospects for job stability and employment growth.

- The regulation of financial products, in particular regarding the derivatives market, can be expected to have ambiguous effects on labor markets. On the one hand, stricter regulation of certain products—regarding, for instance, the disclosure of information, the standardization and trading of these products on centralized platforms, or the outright prohibition of certain products such as uncovered short-selling—is likely to reduce market liquidity, with the risk of higher risk premia and more market volatility, translating into greater macroeconomic uncertainty and lower employment creation. On the other hand, to the extent that these products are themselves at the origin of macroeconomic volatility, regulating such activities can help stabilize the banking sector and thereby lower the macroeconomic risk premia.

## 3. Unemployment Flows and Financial Market Interactions

Based on the original contribution by Wasmer and Weil (2004), in this section, we proceed with the development of our main hypotheses that we want to test in the reminder of this article. In particular, we derive empirically-testable equations for unemployment in- and out-flows that allow a precise quantitative assessment of the interactions of labor market dynamics with the characteristics of financial markets. The section starts by summarizing the main theoretical insights on the interaction between financial markets and labor market outcomes as analyzed by Wasmer and Weil. It then presents

an empirical decomposition of unemployment dynamics that can be used to test the theoretical predicaments using a new database on unemployment flows.

## 3.1. The Impact of Financial Market Frictions on Labor Market Outcomes

Starting from the original contribution to labor market matching of Pissarides (2000), Wasmer and Weil (2004) added financial frictions to the process of job vacancy openings. In their setup, entrepreneurs—prior to opening a job vacancy—need to find an appropriate financier that allows them to finance their capital layouts and the recruitment stage. This happens in a separate step through a frictional matching process on financial markets. Similar to labor markets, the matching process on financial markets can be represented through a (constant-returns-to-scale) matching function with credit liquidity $\phi$ and a repayment rate that is subject to a bilateral Nash bargaining process. Once an entrepreneur has successfully matched a financier, a vacancy can be opened that will be filled at a rate depending on labor market tightness, measured by the ratio of unfilled vacancies, $\mathcal{V}$, over current job seekers, $\mathcal{U}$, in short $\theta \equiv \mathcal{V}/\mathcal{U}$.

The value of such a job vacancy arises from the expected net profit $E\left(\pi\right) = \frac{y-w}{r+\sigma}$, with $y$: output, $w$: worker's salary, $r$: gross interest rate, $\sigma$: match separation rate, minus the recruitment costs, $\gamma\left(\theta\right)$, which are proportional to labor market tightness, $\theta$, or the difficulty to match with a job seeker. In Wasmer and Weil's setup, the value of a vacancy is calculated at the stage when an entrepreneur meets with a prospective bank and will, therefore, depend on the chances of a successful match. In short notation, this yields (see Wasmer and Weil 2004, p. 950):

$$V_t = V\left(\theta_t, r_t, w_t, y, \sigma, \gamma\right) = \frac{q\left(\theta_t\right)}{r_t + q\left(\theta_t\right)} \left[\frac{y - w_t}{r_t + \sigma} - \gamma\left(\theta_t\right)\right] \text{ with } V_\theta, V_r, V_w, V_\sigma, V_\gamma < 0, \ V_y > 0 \quad (1)$$

where $q\left(\theta_t\right)$ describes the matching function with $q_\theta < 0$.

Job creation takes place if the value from a vacant job, $V_t$, exceeds the entry costs for firms and the funding costs for banks. In Wasmer and Weil's model, the joint surplus that arises from a vacancy will be split according to a bargaining rule with the bank's bargaining power measured by $\beta$. Hence, equilibrium arises at the intersection of the schedule of financial and labor market equilibrium:

$$\beta V_t = K\left(\phi_t\right) \text{ and } \left(1 - \beta\right) V_t = C\left(\phi_t\right)$$

where $\phi_t$: available financial funds, $K\left(\phi_t\right)$: fundraising costs, and $C\left(\phi_t\right)$: a firm's entry costs with $K_\phi < 0$, $C_\phi > 0$. In equilibrium, the availability of financial funds will adjust so that:

$$\frac{1 - \beta}{\beta} = \frac{C\left(\bar{\phi}\right)}{K\left(\bar{\phi}\right)}. \quad (2)$$

Besides on financial market tightness, $\phi$, firm entry costs, $C\left(\phi\right)$, are only assumed to depend (negatively) on exogenous barriers to entry as they arise, for instance, from product market regulation. In turn, fundraising costs, $K\left(\phi\right)$, depend (negatively) on banking sector regulation, such as minimum capital requirements, regulatory openness to international capital (in-)flows and the regulation of securities markets. Both fundraising and entry costs of firms will depend on the depth of financial markets that influence the matching process, in particular the speed with which available funds can be matched with entrepreneurial projects. Finally, the equilibrium on financial markets described by Equation (2) depends on the bargaining power of banks, $\beta$, which depend on the structure of the banking sector (number of banks, share of publicly-owned banks), as well as on any interest rate controls.

Similar to Pissarides (2000), Wasmer and Weil's base model kept job destruction exogenous. Our data on labor flows (see Section 4 below) allow, however, distinguishing both margins separately and testing factors that influence job creation and job destruction rates. To derive empirically-testable hypotheses relevant for unemployment inflows, we, therefore, consider that (endogenous) job

destruction takes place whenever the joint surplus of a job, $S_t$, falls below a viability threshold due to the realization of a negative productivity shock. This viability threshold is negotiated prior to the creation of a vacancy and depends on the commitment of banks to provide additional funds in the case of adverse shocks, as well as the quality of the firm's balance sheets (Phelps 1998). As Wasmer and Weil indicated, such commitment might be tested if productivity falls substantially below the zero-profit condition and might depend again on the size and quality of the financial system. Furthermore, and thereby following the earlier literature on endogenous job destruction (see Pissarides 2000; Caballero 2007), we will consider labor market conditions to be relevant for the determination of the optimal threshold at which jobs are being destroyed. In our setup, we assume that the viability condition is defined by the rate of underlying technological progress, given by total factor productivity ($TFP$), whereas realized output per match is given by $y$. In short notation, we, therefore, express the endogenous job destruction as:

$$\sigma_t = \sigma \left( E \left( y < TFP \right) \right)$$

Figure 3 describes how exogenous factors related to financial market development and financial regulation are influencing the position of the two equilibrium schedules and their expected effect on labor market and financial tightness. As shown in Panel A, matching function parameters will not influence the equilibrium position of $\bar{\phi}$ described in Equation (2), and hence, financial development will lead to a proportional shift of both the financial market and labor market equilibrium, leading to less unemployment overall. Stricter banking regulation (Panel B) will only affect financial markets and lead to an upward shift of the BB-schedule, thereby increasing both financial market tightness and unemployment. In contrast, lowering interest rates, for instance through relaxed securities regulation or stiffer banking sector competition, will have ambiguous effects on employment even though it also raises financial market tightness (Panel C). For reasonable parameter values and high unemployment rates, it will actually help expand employment. Finally, lowering entry barriers for firms, for instance the through ease of business regulation, will increase both financial market tightness and employment as only the EE-schedule moves to the right (Panel D).

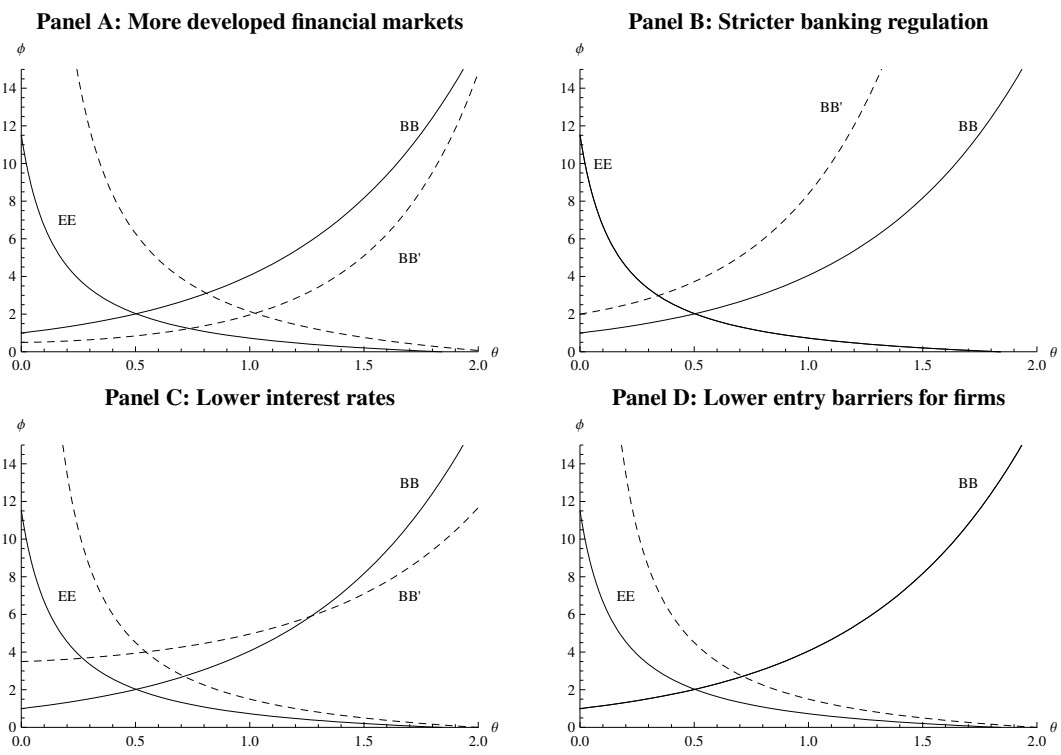

**Figure 3.** Finance and jobs. Note: The EE- and BB-schedules refer to equilibrium on labor and financial markets. (**B**) refers to an increase in $k$.

### 3.2. From Theory to Empirical Analysis: Unemployment Flows Accounting

Testable relations between financial market characteristics and unemployment flows can be set up starting from a labor flow accounting framework. In the following, we concentrate on labor market flows related to flows from employment to unemployment ($IN_t$) and from unemployment into employment ($OUT_t$). These flows can be linked to (absolute) changes in the number of unemployed as follows:

$$\triangle U_t = \triangle L_t - \triangle ET_t = IN_t - OUT_t \tag{3}$$

i.e., the level of unemployment increases with the increase in the labor force, $LF_t$, and decreases with the rise in (total) employment, $ET_t$. Alternatively, unemployment increases when inflows into the unemployment pool, $IN_t$, exceed outflows, $OUT_t$. In order to be operational for our purposes, this flow equation needs to be further refined, taking into account the job creation and destruction process that affects the total amount of jobs available:

$$\triangle ET_t = JobCreation_t - JobDestruction_t \tag{4}$$

i.e., changes in the employment level result from the difference between created vs. destroyed jobs.

Adding our theoretical considerations regarding the impact of financial markets on job dynamics to an earlier empirical formulation of labor market matching models derived by Carlsson et al. (2006), the extensive margin of labor demand can be derived from the surplus value of a vacancy, $V_t$, and is, therefore, determined by a mix of the following factors (see Equation (1) above):

$$JobCreation_t = \alpha_1 + \beta_{11}AD_t + \beta_{12}w_t + \beta_{13}FinDev_t + \beta_{14}r_t + \beta_{15}Share_t \tag{5}$$

where $AD_t = y$: aggregate demand, $w_t$: real wages, $FinDev_t$: financial development/regulation, $r_t$: the real long-term interest rate, $Share_t$: real share price growth. Following considerations by Phelps (1998), we also include balance-sheet effects that arise from variations in (real) share prices and that affect both the rate at which jobs are being created, but more importantly the financial fragility of existing jobs, and hence the rate of job destruction.

Similarly, job destruction will be affected by the rate of technological progress, the real interest rate (through the discounted future benefits of an ongoing relationship), import competition, wages, and aggregate demand:

$$JobDestruction_t = \alpha_2 + \beta_{21}TFP_t + \beta_{22}r_t + \beta_{23}FinDev_t + \beta_{24}w_t + \beta_{25}AD_t + \beta_{26}Share_t \tag{6}$$

where $TFP_t$: an indicator for total factor productivity, $r_t$: the real (long-term) interest rate, $FinDev_t$: financial development/regulation.

Finally, unemployment dynamics as described above are also affected by changes in the labor force. We build upon standard theories about the determinants of labor supply by considering the following equation for changes in labor force growth (see, for instance, Burniaux et al. 2003):

$$\triangle L_t = \alpha_3 + \beta_{31}\triangle L_{t-1} + \beta_{32}\triangle u_{t-1} + \beta_{33}Tax_t \tag{7}$$

where $\beta_{32}$ represents the discouraged worker effect, which depresses the labor force (with an expected negative sign).

The five Equations (3)–(7) form the basis of our labor market flow model. Due to the lack of internationally-comparable data on job creation and destruction rates, however, the model needs to be rewritten to match with our database. This can be done by bringing in accordance job creation rates with unemployment outflows, on the one hand, and job destruction with unemployment inflows, on the other. This requires that the determinants of labor supply as specified in Equation (7) are plugged into the appropriate unemployment flow equation. Indeed, unemployment inflows and outflows do not match exactly job destruction and job creation. Some unemployment inflow happens from

inactivity, while some of those loosing their job might drop out immediately to inactivity if they do not qualify for any benefits. Similarly, job creation can happen out of inactivity (for instance, through self-employment), while some people might "flow out of" unemployment at the end of their benefit period and into inactivity. As a consequence, using unemployment flows instead of job creation and destruction rates might overestimate employment dynamics due to the failure to take out flows back and forth from and to inactivity. It might also overestimate the variation of employment growth when the inactivity rate fluctuates with the business cycle (as is suggested by the discouraged worker effect).

In our specification, we consider that the discouraged worker effect will create additional unemployment outflows. On the other hand, increasing supply in the available workforce will show up as additional unemployment inflows. Tax-related changes in labor supply are considered to affect both unemployment inflows and outflows. Besides these adjustments to our specification, we consider both unemployment inflows and outflows to follow dynamic adjustment processes, instead of estimating them at levels. This way, we cope with systematic under- or over-estimation of flows over the cycle that are due to linkages between unemployment and inactivity. In addition, by considering contemporaneous interactions between the two flow directions, we also take care of the possibility that we are overestimating the impact of unemployment flows on employment variation: higher contemporaneous inflows will also increase outflows as part of it goes into inactivity. Similarly, higher outflows might partly imply an increase in inactivity that will show up in increased inflow rates. We will therefore estimate the following two equations related to unemployment dynamics:

$$OUT_t \;=\; \alpha_{OUT} + \tilde{\beta}_{11} OUT_{t-1} + \tilde{\beta}_{12} X_t^{JobCreation} + \tilde{\beta}_{13} \triangle u_{t-1} \tag{8a}$$

$$IN_t \;=\; \alpha_{IN} + \tilde{\beta}_{21} IN_{t-1} + \tilde{\beta}_{22} X_t^{JobDestruction} + \tilde{\beta}_{23} \triangle L_{t-1} \tag{8b}$$

where $X_t^{JobCreation}$ and $X_t^{JobDestruction}$ correspond to the different explanatory variables in Equations (5) and (6), respectively. Equations (8a) and (8b) will form the base model for the following extensions of our labor flow model.

## 4. Data, Methodology, and Hypotheses

### 4.1. Data: Unemployment Flows, Labor Market Institutions, and Financial Structure

The paper brings together three databases: unemployment in- and outflows, general macroeconomic data, and financial market dynamics. The resulting database covers 20 OECD countries over most of the period between 1970 and 2009 on an annual basis (time coverage changes depending on the particular specification used). Even though information on both unemployment flows and financial market development is available for more recent periods, we restrict the sample on purpose to allow comparability with the IMF's financial market reform index, which is only available until 2005. This way, we interpret our scenario results as the impact of financial market regulation on job dynamics had the regulation been in place prior to the global financial crisis.

Unemployment flows are taken from the International Labour Organization (ILO) database on unemployment flows. This database follows the methodology developed by Shimer (2005, 2012) and Elsby et al. (2013). The data are constructed on the basis of OECD information regarding unemployment stocks and unemployment duration of different duration lengths. In contrast to similar information provided by Shimer (2012) or Petrongolo and Pissarides (2008), our data on unemployment flows follow the methodology suggested by Elsby et al. (2013) that allows for a systematic cross-country analysis. In our case, we take advantage of the larger country coverage to use the increased number of degrees of freedom (within a panel-data context) in order to test for a larger number of determinants of unemployment flows.

Information on different aspects of financial market development is based on the updated version of the database originally provided by Beck et al. (2000). The database contains information on the asset and liability side of the financial sector (banking, bond, and stock markets), as well as

diverse indicators regarding the performance of the banking market (return-on-assets, return-on-equity, net interest margin, concentration rate). In the estimations below, we have been concentrating on liquidity measures to reflect the quantitative impact that government net lending might have on available loan-able funds.

Indicators on financial sector reforms in OECD countries since the 1970s have been developed by Abiad et al. (2008). The database establishes an overall indicator on the (time-varying) state of financial market regulation. In addition, the database allows distinguishing between different areas of regulation, including interest rate ceilings, credit growth restrictions, entry barriers for banks, the extent of (financial sector) privatizations, and security market reforms. In our regressions, we make use of the most detailed level of information, although we only report coefficients when they satisfy a minimum degree of statistical significance.

As the financial market regulation index produced by Abiad et al. (2008) is only available until 2005, we make use of additional information on the evolution of financial market regulation provided by the Heritage Foundation. The Heritage Index of Economic Freedom assesses cross-country differences of market regulation and state interventions in different areas since 1995 and up to 2018 (see https://www.heritage.org/index/). For our purposes, six sub-indicators of the overall index have been used: financial freedom, investment freedom, trade freedom, monetary freedom, level of government spending, and business freedom. We used simple panel data regressions to make out-of-sample imputations, which allowed an expansion of the original financial reform database to up to 32 OECD countries covering all years from 1995–2018 (detailed regression output of the imputation equations is available from the author upon request).

Information on international capital flows has been used from Lane and Milesi-Ferretti (2007). The database contains information regarding internationally-traded assets and liabilities, such as foreign direct investment, portfolio equity and debt flows, and financial derivatives. Both share indicators and (real) growth rates (deflated using the GDP deflator) have been used in this paper.

The database has been complemented with macro-economic and labor market information taken from the OECD Economic Outlook database and the OECD Main Economic Indicators. In particular, data regarding total employment and labor force developments, capital stock estimates, and interest rates are taken from there. In addition, an indicator of real share price increases has been developed on the basis of OECD information, using the GDP deflator to deflate nominal share prices.

### 4.2. Methodology and Hypotheses

To exploit the cross-country nature of our data, we apply standard panel data techniques. The theoretical equations developed in Section 3 are both formulated in level terms, which would suggest the use of simple OLS to test the different determinants of unemployment flows. Three issues arise, however, in this respect:

- Country-specific information is not always available for all variables over the entire time period. Moreover, some of the financial reform indicators suffer from very limited time variability within country panels.
- Both unemployment in- and out-flows are highly persistent within countries, introducing problems of auto-correlation.
- Some of the right-hand side variables are endogenous to the dependent variable (in- and out-flows).

To address these problems, we follow Beck et al. (2000) and use the Arellano–Bond (AB) system General Method of Moments (GMM)estimator, taking into account the low-order moving-average correlation of the error terms as we have relatively long time series. As the AB-GMM estimator is known to be inconsistent for panels with a large time dimension due to the proliferation of instruments, we limit the number of instruments to satisfy the Sargan over-identification test in line with standard practice in the finance-and-growth literature (see Beck et al. 2000).

In our base specification, we rely on the setup identified by Ernst (2011) and Ernst and Rani (2011). In order to identify the interactions with financial markets as discussed above, we separately introduce variables related to financial development and financial market regulation to avoid problems of multi-collinearity (see Sections 5.1 and 5.2). In Section 5.3, we use our imputed financial reform index as a robustness check to our baseline regressions. In a separate step, we simulate a joint model of financial determinants of unemployment outflows and inflows in order to present the labor market outcomes of reform proposals that are currently being discussed or in the process of being implemented (Section 6).

Based on our theoretical considerations in Section 3, we want to contribute to a better understanding of labor market dynamics as financial markets develop:

1. Financial market development might simultaneously increase job creation and lower job destruction due to stronger investment that is being directed towards its most productive use;
2. financial market development might also lead to higher labor market turnover as both unemployment out- and in-flows increase due to faster restructuring, which is implied by deeper financial markets;
3. in contrast, financial market development might depress both job creation and increase job destruction if it brings about an increase in unproductive, speculative activities that do not generate jobs.

As regards financial market regulation and based on the macro-finance literature discussed above, two main hypotheses may be put forward:

1. Stricter regulation of banking sector activities will lead to an increase in the user cost of capital, thereby reducing unemployment outflows and increasing unemployment inflows.
2. Better prudential regulation reduces financial market stress and stabilizes the banking sector, thereby relieving credit constraints; this should support job creation and limit job destruction.

In the next section, we will present the empirical results to see whether they confirm our hypotheses.

## 5. Financial Market Determinants of Unemployment Flows

This section presents evidence regarding the different hypotheses on the relationship between financial markets and unemployment flows. First, we will look into the influence of the size of the financial sector and the importance of international financial flows on labor market dynamics. In Section 5.2, our interest then turns to analyzing the effects of financial market regulation on unemployment flows.

### 5.1. Financial Market Development and Unemployment Flows

Tables 1 and 2 summarize the results regarding the impact of various facets of financial market development on unemployment flows. Different aspects are covered, ranging from financial market development as measured by the share of private credit in GDP (equations (1.1) and (2.1)) over international financial integration (equations (1.2) and (2.2)), to the importance of stock and private bond markets (equations (1.3), (1.4), (2.3), and (2.4)). A fifth equation is added, estimating the joint effect of these four aspects to assess the extent to which these four indicators cover different aspects of financial market development.

Our results confirm the original insight of Wasmer and Weil: increased financial market development as measured by a higher credit-to-GDP ratio lowers unemployment by lowering unemployment inflows (equation (1.1)) and increasing unemployment outflows (equation (2.1). This is true even when controls for other forms of financial market development are being introduced (see equations (1.5) and (2.5)).

However, such unambiguous results do not apply to other financial market indicators. When considering international financial openness as measured by international capital flows as

a share of GDP, unemployment inflows are being reduced, but so are unemployment outflows. This suggests that rather than stimulate an economy's restructuring, such capital inflows tend to make unemployment dynamics less volatile, but also less conducive to economic restructuring.

In contrast, both more developed stock and bond markets yield higher labor market turbulence. Both indicators are associated with higher unemployment inflows and higher unemployment outflows. This suggests that market-based financial development—rather than credit/bank-based finance—is likely to be more conducive to labor market restructuring and eventually to a more rapid adaptation of labor markets to external shocks. Whether the overall contribution of either stock market or bond market development leads to higher or lower employment levels seems to be a question of the particular specification used. In particular for the indicator of stock market development, estimated coefficients seem to depend largely on the specification used (compare equations (1.3) and (1.5) in Table 1), as well as on the number of countries retained in each sample.

**Table 1.** Financial market determinants of unemployment inflows.

| | Dependent Variable: Unemployment Inflows | | | | |
| | (1.1) | (1.2) | (1.3) | (1.4) | (1.5) |
|---|---|---|---|---|---|
| Unemployment inflows (lagged) | 0.881 *** $(1.9 \times 10^{-2})$ | 0.912 *** $(1.9 \times 10^{-2})$ | 0.836 *** $(2.2 \times 10^{-2})$ | 0.831 *** $(2.4 \times 10^{-2})$ | 0.808 *** $(3.1 \times 10^{-2})$ |
| Output gap | $-3.3 \times 10^{-2}$ *** $(9.0 \times 10^{-3})$ | $-2.6 \times 10^{-2}$ ** $(1.1 \times 10^{-2})$ | $-3.8 \times 10^{-2}$ *** $(1.0 \times 10^{-2})$ | $-4.2 \times 10^{-2}$ *** $(1.0 \times 10^{-2})$ | $-5.1 \times 10^{-2}$ *** $(1.5 \times 10^{-2})$ |
| TFP growth (lagged) | $2.4 \times 10^{-1}$** $(1.0 \times 10^{-1})$ | $2.8 \times 10^{-1}$ *** $(1.2 \times 10^{-1})$ | $2.6 \times 10^{-1}$ ** $(1.1 \times 10^{-1})$ | $2.5 \times 10^{-1}$ ** $(1.2 \times 10^{-2})$ | $3.4 \times 10^{-1}$ ** $(1.6 \times 10^{-1})$ |
| Labor force growth (lagged) | 1.247 *** $(3.2 \times 10^{-1})$ | $-1.4 \times 10^{-2}$ $(1.5 \times 10^{-2})$ | 1.228 *** $(3.0 \times 10^{-1})$ | 1.291 *** $(3.2 \times 10^{-1})$ | $8.7 \times 10^{-1}$ *** $(2.8 \times 10^{-1})$ |
| Real share price growth (lagged) | $-1.3 \times 10^{-1}$ *** $(3.7 \times 10^{-2})$ | $-1.5 \times 10^{-1}$ *** $(3.9 \times 10^{-2})$ | $-9.8 \times 10^{-2}$ *** $(3.8 \times 10^{-2})$ | $-1.2 \times 10^{-1}$ *** $(3.9 \times 10^{-2})$ | $-1.9 \times 10^{-1}$ *** $(4.9 \times 10^{-2})$ |
| Financial market development | $-4.5 \times 10^{-2}$ ** $(2.1 \times 10^{-2})$ | | | | $-1.2 \times 10^{-1}$ *** $(3.2 \times 10^{-2})$ |
| International financial openness | | $-1.1 \times 10^{-2}$ ** $(4.0 \times 10^{-3})$ | | | $-1.8 \times 10^{-2}$ *** $(5.6 \times 10^{-3})$ |
| Stock market development | | | $1.9 \times 10^{-2}$ ** $(1.0 \times 10^{-2})$ | | $8.2 \times 10^{-2}$ *** $(2.9 \times 10^{-2})$ |
| Private bond market capitalization | | | | $1.5 \times 10^{-1}$ *** $(3.1 \times 10^{-2})$ | $2.0 \times 10^{-1}$ *** $(4.8 \times 10^{-2})$ |
| Observations | 399 | 315 | 342 | 307 | 206 |
| Number of countries | 19 | 19 | 19 | 18 | 18 |

Note: TFP: total factor productivity. All estimates performed using the Arellano–Bond system GMMestimator. Standard errors in parentheses. All regressions pass the Sargan over-identification test to validate the number of instruments. The number of asterisks indicates the statistical significance level: *** $p < 0.01$, ** $p < 0.05$, * $p < 0.1$.

**Table 2.** Financial market determinants of unemployment outflows.

| | Dependent Variable: Unemployment Outflows | | | | |
| --- | --- | --- | --- | --- | --- |
| | **(2.1)** | **(2.2)** | **(2.3)** | **(2.4)** | **(2.5)** |
| Unemployment outflows (lagged) | 0.770 *** $(2.9 \times 10^{-2})$ | 0.726 *** $(3.5 \times 10^{-2})$ | 0.770 *** $(3.2 \times 10^{-2})$ | 0.712 *** $(3.8 \times 10^{-2})$ | 0.700 *** $(4.4 \times 10^{-2})$ |
| Output gap | $1.9 \times 10^{-2}$ *** $(7.0 \times 10^{-3})$ | $2.4 \times 10^{-2}$ *** $(9.0 \times 10^{-3})$ | $1.1 \times 10^{-2}$ $(7.0 \times 10^{-3})$ | $1.6 \times 10^{-2}$ * $(9.0 \times 10^{-3})$ | $3.5 \times 10^{-2}$ ** $(1.1 \times 10^{-2})$ |
| Real long term interest rate | $-4.0 \times 10^{-3}$ $(6.0 \times 10^{-3})$ | $-1.6 \times 10^{-2}$ ** $(7.0 \times 10^{-3})$ | $1.0 \times 10^{-3}$ $(6.0 \times 10^{-3})$ | $-3.0 \times 10^{-3}$ *** $(8.0 \times 10^{-3})$ | |
| Wage-interest rate ratio | $-4.2 \times 10^{-2}$ ** $(2.0 \times 10^{-2})$ | $-8.8 \times 10^{-2}$ *** $(4.5 \times 10^{-2})$ | $-5.9 \times 10^{-2}$ *** $(2.1 \times 10^{-2})$ | $-5.4 \times 10^{-2}$ ** $(2.1 \times 10^{-2})$ | $-1.0 \times 10^{-1}$ * $(5.2 \times 10^{-2})$ |
| Real share price growth | | | | | $1.7 \times 10^{-1}$ ** $(7.0 \times 10^{-2})$ |
| Gross fixed capital investment | 1.232 *** $(2.0 \times 10^{-1})$ | 1.188 *** $(2.4 \times 10^{-1})$ | 1.164 *** $(2.1 \times 10^{-1})$ | 1.288 *** $(2.4 \times 10^{-1})$ | |
| Financial market development | 0.149 *** $(3.1 \times 10^{-2})$ | | | | $8.1 \times 10^{-2}$ ** $(4.1 \times 10^{-2})$ |
| International financial openness | | $-0.044$ *** $(8.0 \times 10^{-3})$ | | | $-4\text{-}5 \times 10^{-2}$ *** $(8.4 \times 10^{-3})$ |
| Stock market development | | | 0.182 *** $(3.7 \times 10^{-2})$ | | 0.171 *** $(4.8 \times 10^{-2})$ |
| Private bond market capitalization | | | | 0.378 *** $(6.2 \times 10^{-2})$ | 0.358 *** $(7.2 \times 10^{-2})$ |
| Observations | 390 | 302 | 332 | 303 | 214 |
| Number of countries | 20 | 20 | 20 | 18 | 18 |

Note: All estimates performed using the Arellano–Bond system GMM estimator. Standard errors in parentheses. All regressions pass the Sargan over-identification test to validate the number of instruments. The number of asterisks indicates the statistical significance level: *** $p < 0.01$, ** $p < 0.05$, * $p < 0.1$.

## 5.2. Financial Market Reforms and Labor Market Flows

The theoretical model discussed in Section 3 also offers the possibility to analyze directly the effect of financial sector reforms on job dynamics. Since the financial market crisis most of the discussion in this area has focused on the impact of reforms on financial sector stability and diversification rather than on real economic growth, with some notable exceptions (see, for instance, Admati et al. 2013, Deli and Hasan 2018, Kashyap et al. 2010, Basel Committee on Banking Supervision 2010, Financial Stability Board 2010). Rare have been the papers that have specifically looked into the effects such reforms might have on labor markets (see, for instance, Shapiro and Gómez 2015).

Using the same methodology as the one outlined in Section 3, we use information recently made available on financial reforms (see Section 4, which discusses the data). The available information stops in 2005 and covers less countries than the one on financial market development, but does still allow panel data analysis with a medium-sized panel. The results of these estimates are presented in Table 3 for unemployment outflows and in Table 4 for unemployment inflows.

The results on unemployment outflows are broadly in line with those uncovered regarding the effects of financial market development on job creation. Removing credit provisions or loosening credit controls impact unemployment outflows positively, reflecting the fact that stronger credit growth pushes up employment creation. Similarly, reforming securities markets will also help increase job creation rates. On the other hand, loosening interest rate controls does not seem to have a significant effect, reflecting the ambiguity of lower interest rates on labor market liquidity as discussed in the theoretical section.

**Table 3.** Financial sector reforms and unemployment outflows.

| | (3.1) | (3.2) | (3.3) | (3.4) | (3.5) | (3.6) | (3.7) | (3.8) | (3.9) | (3.10) | (3.11) |
|---|---|---|---|---|---|---|---|---|---|---|---|
| | | | | | **Dependent Variable: Unemployment Outflows** | | | | | | |
| Unemployment outflows (lagged) | 0.734 *** | 0.799 *** | 0.800 *** | 0.745 *** | 0.788 *** | 0.836 *** | 0.836 *** | 0.815 *** | 0.816 *** | 0.838 *** | 0.857 *** |
| | (0.0345) | (0.0283) | (0.0283) | (0.0342) | (0.0320) | (0.0273) | (0.0307) | (0.0275) | (0.0301) | (0.0271) | (0.0273) |
| Output gap | 0.0133 ** | 0.00451 | 0.00498 | 0.0134** | 0.0103* | 0.0139** | 0.00938 | 0.00637 | 0.00354 | 0.00912 | 0.00283 |
| | (0.00593) | (0.00626) | (0.00626) | (0.00628) | (0.00624) | (0.00662) | (0.00639) | (0.00621) | (0.00651) | (0.00621) | (0.00670) |
| Real short-term interest rate | −0.0196 *** | −0.0133 *** | −0.0136 *** | −0.0189 *** | −0.0143 *** | −0.0126 ** | −0.0200 *** | −0.00946 * | −0.0154 *** | −0.0119 ** | −0.0124 ** |
| | (0.00478) | (0.00505) | (0.00506) | (0.00483) | (0.00522) | (0.00551) | (0.00527) | (0.00536) | (0.00510) | (0.00522) | (0.00542) |
| Change in wage-capital ratio | −0.105 *** | −0.132 *** | −0.132 *** | −0.108 *** | −0.116 *** | −0.107 *** | −0.121 *** | −0.129 *** | −0.131 *** | −0.121 *** | −0.135 *** |
| | (0.0316) | (0.0332) | (0.0333) | (0.0321) | (0.0330) | (0.0345) | (0.0362) | (0.0333) | (0.0336) | (0.0332) | (0.0364) |
| Real share price inflation | 0.102 ** | 0.0816 * | 0.0846 * | 0.104 ** | 0.109 ** | 0.118 ** | 0.111 ** | 0.104 ** | 0.145 *** | 0.103 ** | 0.136 *** |
| | (0.0466) | (0.0496) | (0.0497) | (0.0473) | (0.0482) | (0.0505) | (0.0510) | (0.0492) | (0.0503) | (0.0492) | (0.0525) |
| Gross fixed capital formation | 4.383 *** | 5.038 *** | 5.009 *** | 4.145 *** | 4.302 *** | 2.803 *** | 3.752 *** | 4.582 *** | 4.762 *** | 3.912 *** | 4.575 *** |
| | (0.874) | (0.954) | (0.956) | (0.939) | (0.913) | (0.949) | (0.979) | (0.933) | (0.963) | (0.907) | (1.015) |
| Growth in real household disposable income | 1.361 ** | 1.698 *** | 1.702 *** | 1.375 ** | 1.481 ** | 1.400 ** | 1.348 ** | 1.592 ** | 1.859 *** | 1.523 ** | 1.851 *** |
| | (0.604) | (0.639) | (0.640) | (0.613) | (0.626) | (0.656) | (0.670) | (0.638) | (0.654) | (0.637) | (0.685) |
| Financial derivatives liabilities (in % of GDP, lagged) | −2.760 *** | −2.883 *** | −2.891 *** | −2.665 *** | −2.174 *** | −3.205 *** | −2.538 *** | −3.351 *** | −2.237 *** | −2.729 *** | −2.851 *** |
| | (0.552) | (0.585) | (0.588) | (0.557) | (0.525) | (0.702) | (0.695) | (0.620) | (0.567) | (0.587) | (0.711) |
| Removing directed credit provisions (lagged) | | 0.130 *** | | | | | | | | | |
| | | (0.0175) | | | | | | | | | |
| Loosening of credit controls (lagged) | | | 0.130 *** | | | | | | | | |
| | | | (0.0177) | | | | | | | | |
| Loosening of interest rate controls (lagged) | | | | 0.0168 | | | | | | | |
| | | | | (0.0214) | | | | | | | |
| Lifting of entry barriers (lagged) | | | | | 0.00438 | | | | | | |
| | | | | | (0.0106) | | | | | | |
| Bank privatization (lagged) | | | | | | 0.0810 *** | | | | | |
| | | | | | | (0.0189) | | | | | |
| Capital account openness (lagged) | | | | | | | 0.0514 *** | | | | 0.0350 ** |
| | | | | | | | (0.0168) | | | | (0.0156) |
| Financial reforms (lagged) | | | | | | | | 0.0251 *** | | | |
| | | | | | | | | (0.00373) | | | |
| Securities markets' deregulation (lagged) | | | | | | | | | 0.203 *** | | 0.123 *** |
| | | | | | | | | | (0.0327) | | (0.0345) |
| Prudential regulation of banks (lagged) | | | | | | | | | | 0.0566 *** | 0.0376 *** |
| | | | | | | | | | | (0.00879) | (0.00911) |
| Constant | −0.531 *** | −0.805 *** | −0.803 *** | −0.555 *** | −0.474 *** | −0.514 *** | −0.458 *** | −0.886 *** | −1.036 *** | −0.480 *** | −0.883 *** |
| | (0.0727) | (0.106) | (0.107) | (0.110) | (0.0730) | (0.0915) | (0.0975) | (0.121) | (0.147) | (0.0741) | (0.152) |
| Observations | 152 | 152 | 152 | 152 | 152 | 152 | 145 | 152 | 152 | 152 | 145 |
| Number of countries | 8 | 8 | 8 | 8 | 8 | 8 | 8 | 8 | 8 | 8 | 8 |

Note: All estimates performed using the Arellano–Bond system GMM estimator. Standard errors in parentheses. All regressions pass the Sargan over-identification test to validate the number of instruments. The number of asterisks indicates the statistical significance level: *** $p < 0.01$, ** $p < 0.05$, * $p < 0.1$.

**Table 4.** Financial sector reforms and unemployment inflows.

| | (4.1) | (4.2) | (4.3) | (4.4) | (4.5) | (4.6) | (4.7) | (4.8) | (4.9) |
|---|---|---|---|---|---|---|---|---|---|
| | | | | | Dependent Variable: Unemployment Inflows | | | | |
| Unemployment inflows (lagged) | 0.940 *** (0.0179) | 0.942 *** (0.0229) | 0.911 *** (0.0235) | 0.941 *** (0.0176) | 0.930 *** (0.0177) | 0.925 *** (0.0170) | 0.935 *** (0.0164) | 0.912 *** (0.0177) | 0.939 *** (0.0184) |
| Output gap | −0.00208 (0.00288) | −0.00654 (0.00548) | −0.00632 (0.00514) | −0.00194 (0.00289) | −0.00237 (0.00287) | −0.00286 (0.00286) | −0.00244 (0.00287) | −0.00284 (0.00281) | −0.00813 *** (0.00303) |
| Total factor productivity (annual growth) | 0.167 ** (0.0766) | 0.421 *** (0.112) | 0.442 *** (0.113) | 0.169 ** (0.0765) | 0.170 ** (0.0756) | 0.177 ** (0.0750) | 0.163 ** (0.0763) | 0.171 ** (0.0775) | 0.202 ** (0.0799) |
| Labor force participation rate (lagged) | 0.495 ** (0.233) | −0.0287 (0.280) | 0.229 (0.283) | 0.471 ** (0.231) | 0.605 *** (0.215) | 0.617 *** (0.223) | 0.494 ** (0.210) | 0.492 ** (0.207) | 0.849 *** (0.223) |
| User cost of capital (lagged) | 0.01000 *** (0.00335) | 0.0173 *** (0.00509) | 0.0194 *** (0.00500) | 0.0100 *** (0.00334) | 0.0100 *** (0.00326) | 0.00914 *** (0.00325) | 0.00964 *** (0.00327) | 0.00967 *** (0.00328) | 0.00785 ** (0.00344) |
| Real imports growth (lagged) | −0.518 *** (0.112) | −0.432 ** (0.180) | −0.417 ** (0.170) | −0.517 *** (0.112) | −0.506 *** (0.111) | −0.502 *** (0.111) | −0.517 *** (0.111) | −0.496 *** (0.109) | −0.392 *** (0.113) |
| Removing directed credit provisions (lagged) | 0.00526 (0.00983) | | | | | | | | |
| Removing credit ceilings (lagged) | | −0.0530 (0.0583) | −0.0465 (0.0551) | | | | | | |
| Financial derivatives liabilities (in % of GDP, lagged) | | | −5.842 *** (2.063) | | | | | −1.118 *** (0.372) | 0.194 (0.406) |
| Loosening of credit controls (lagged) | | | | 0.00736 (0.0106) | | | | | |
| Loosening of interest rate controls (lagged) | | | | | −0.0109 (0.0114) | | | | |
| Lifting of entry barriers (lagged) | | | | | | −0.0130* (0.00683) | | | |
| Bank privatization (lagged) | | | | | | | 0.0121 (0.00751) | 0.0250 *** (0.00844) | |
| Capital account openness (lagged) | | | | | | | | | −0.0530 *** (0.00939) |
| Securities markets' deregulation (lagged) | | | | | | | | | 0.0546 *** (0.0152) |
| Prudential regulation of banks (lagged) | | | | | | | | | −0.0123 ** (0.00547) |
| Constant | −0.767 *** (0.178) | −0.566 *** (0.201) | −0.884 *** (0.213) | −0.752 *** (0.176) | −0.848 *** (0.178) | −0.888 *** (0.184) | −0.797 *** (0.177) | −0.933 *** (0.181) | −1.064 *** (0.175) |
| Observations | 221 | 75 | 75 | 221 | 221 | 221 | 221 | 221 | 211 |
| Number of countries | 12 | 5 | 5 | 12 | 12 | 12 | 12 | 12 | 12 |

Note: All estimates performed using the Arellano–Bond system GMM estimator. Standard errors in parentheses. All regressions pass the Sargan over-identification test to validate the number of instruments. The number of asterisks indicates the statistical significance level: *** $p < 0.01$, ** $p < 0.05$, * $p < 0.1$.

Table 3 also offers two noticeable differences with respect to results presented in earlier sections. First, increasing capital account openness seems to contribute positively to job creation rates, whereas de facto financial openness had depressed it (see Table 2, equations (3.2) and (3.5)). This might have to do with the fact that other reforms are typically being undertaken simultaneously, which helps overall job creation to be stimulated even though the direct effect from higher financial integration is negative. Second, stricter prudential regulation leads to stronger unemployment outflows, in contrast to what the theoretical results presented in Section 3 would have suggested. This might have to do with the fact that such reforms lead to lower volatility of the real economy with less frequent cycles that have a lower amplitude, which both should be expected to influence job creation positively. This effect, however, has not properly been taken care of in the theoretical model underlying Section 3, which abstracts from any effects from financial instability.

Similar to unemployment outflows, the results on the impact of financial regulation on unemployment inflows presented in Table 4 are also in line with the earlier results of Table 1, at least as regards reforms to securities market regulation. Most of the indicators on reforming credit provisions and interest rate controls are insignificant, suggesting that they have little impact on the separation rate. The measured impact of credit growth on unemployment inflows might, therefore, stem from an overall aggregate demand effect rather than from a specifically finance-related impact of credit on job separation. As before, capital account openness has a significant effect that is different from the observed effect of de facto international financial integration; taken together, these results suggest that de jure financial openness is unambiguously positively related to employment growth. Furthermore, prudential regulation of banks supports lower job separation rates, yielding an overall positive contribution of prudential regulation on employment growth, again a result that is somewhat at odds with the theoretical predictions of Section 3.

With these results of the impact of individual financial market reforms on labor market outcomes, we are now ready to turn to analyze broader implications of across-the-board financial sector reform as it is currently being debated. This will be done in the following section.

*5.3. Robustness Check*

As the financial reform index developed by Abiad et al. (2008) is only available until 2005, we extend our reform database using the methodology described in Section 4. In particular, we use six sub-indicators of the Heritage Foundation Index of Economic Freedom in order to expand the reform index beyond 2005. As Figure 4 displays, the extended reform indicator suggests some reversal of financial market deregulation in the aftermath of the global financial crisis. Nevertheless, financial re-regulation never returned to the level of strictness and intervention as observed in the 1980s. Even the most strictly regulated financial market in the 2010s was only marginally more so than in the second half of the 1990s. What is more, by 2018, many countries had already reverted back to a more relaxed stance on financial market regulation. The imputed indicator gives, indeed, a good reflection of the debate in the literature as reviewed in Section 2, which highlighted the on-going need for financial market regulation and the worry that ten years after the crisis, many lessons might have been lost already (Tooze 2018).

Applying this extended financial reform index to the specifications for unemployment inflows and outflows as described in Section 4, Tables 5 and 6 summarize the main results of the estimations. As before, the Sargan over-identification test is passed in all specifications, indicating that the number of instruments used does not lead to overconfidence of the estimated parameters. However, given that the extension of the financial reform index is based on imputation techniques rather than by assessing actual regulatory changes, the following estimations should be considered as robustness checks only.

Confirming a result from the previous section, using the extended financial reform indicator displays an increase in labor market turbulence with a rise in both unemployment inflows and outflows (Specifications (5.1) and (6.1)). A major component in this seems to the opening of the capital account (Specifications (5.2), (6.2), (5.6) and (6.6)), which raises both in- and out-flows. In this regard,

the extended indicator seems to suggest a change in the impact of capital account opening, as in the specification looking only at the impact up to 2005, capital account opening was associated with a reduction in unemployment inflows (see Table 4, equation (4.9)). Again, this ties in well with the shift in perception in the literature that has moved to a more cautious approach as regards international capital market deregulation. Securities market deregulation, on the other hand, is associated with ambiguous effects on unemployment outflows (depending on the specification used) and no longer showed any significant correlation with unemployment inflows.

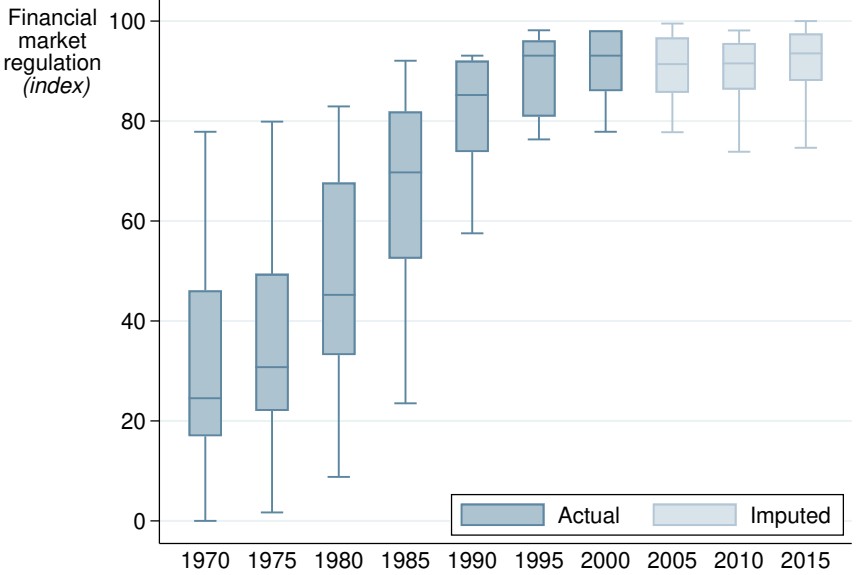

**Figure 4.** Financial reforms: extended indicator for OECD countries. Note: The chart displays cross-country variation of the IMF financial reform index for OECD countries between 1973 and 2018 in five-year averages. Values for years after 2005 are imputed, using the method described in the text. The IMF financial reform index has been rescaled to range form zero (no reforms) to 100 (full liberalization).

(Re-)regulation of the banking sector is partly confirmed by the extended reform indicator. Notably, lifting of entry barriers for banks does not seem to bring benefits to the labor markets, as both unemployment inflows and outflows decline, suggesting an overall ambiguous impact on employment growth. This is in line with findings reported above for the shorter sample prior to 2005. Deregulation of credit controls and credit ceilings also confirm the findings from the shorter sample, albeit at relatively low statistical significance levels. Looking at the extended indicator, prudential regulation of banks seems to bring less benefits than in the shorter sample. While still yielding improvements in unemployment outflows, it also leads to increases in unemployment inflows, indicating an overall increase in labor market turbulence, in contrast to findings in the previous section. We will see, however, in the scenario analysis that once a larger specification is considered that includes also indication of the prevalence of financial crisis, such a result can also be found when restricting the sample to pre-crisis observations.

Overall, the robustness check using the extended financial reform indicator suggests the substantial impact of financial market deregulation on labor market turbulence, with ambiguous effects on the overall level of employment growth. The careful combination of reforms and re-regulation of certain areas of financial markets, notably as regards capital accounts, promise to yield positive impacts on labor markets by both strengthening competitive forces in the banking sector and reducing the incidence of financial crisis and turbulence. The following section will explore such reform packages in more detail, using the shorter sample to provide an assessment as to what would have happened had these reform packages been in place prior to the GFC.

**Table 5.** Robustness check: financial market reforms and unemployment inflows.

| | Dependent Variable: Unemployment Inflows | | | | | | |
|---|---|---|---|---|---|---|---|
| | (5.1) | (5.2) | (5.3) | (5.4) | (5.5) | (5.6) | (5.7) |
| Inflows *(lagged)* | 0.762 *** (0.0275) | 0.767 *** (0.0279) | 0.770 *** (0.0287) | 0.795 *** (0.0268) | 0.790 *** (0.0266) | 0.735 *** (0.0290) | 0.757 *** (0.0322) |
| Output gap | −0.0194 *** (0.00336) | −0.0187 *** (0.00331) | −0.0177 *** (0.00345) | -0.0197 *** (0.00311) | −0.0206 *** (0.00314) | −0.0170 *** (0.00334) | −0.0160 *** (0.00376) |
| Total factor productivity growth | $9.85 \times 10^{-5}$ *** ($2.20 \times 10^{-5}$) | $8.64 \times 10^{-5}$ *** ($2.11 \times 10^{-5}$) | $7.99 \times 10^{-5}$ *** ($2.16 \times 10^{-5}$) | $8.12 \times 10^{-5}$ *** ($2.15 \times 10^{-5}$) | $8.31 \times 10^{-5}$ *** ($2.16 \times 10^{-5}$) | $9.77 \times 10^{-5}$ *** ($2.18 \times 10^{-5}$) | $8.04 \times 10^{-5}$ *** ($2.27 \times 10^{-5}$) |
| Real long term interest rate | 0.0143 *** (0.00509) | 0.0102 ** (0.00466) | 0.0134 ** (0.00526) | | | 0.0136 *** (0.00522) | 0.0128 ** (0.00607) |
| Real long term interest rate (change) | | | | 0.0131 *** (0.00463) | 0.0137 *** (0.00464) | | |
| Real wage growth *(lagged)* | 0.00902 ** (0.00403) | 0.00644 * (0.00389) | 0.00694 * (0.00404) | 0.00538 * (0.00296) | 0.00739 ** (0.00297) | 0.00680 * (0.00401) | 0.00785 * (0.00451) |
| Financial reforms *(lagged)* | 0.0214 *** (0.00739) | | | | | | |
| Capital account opening | | 0.178 *** (0.0446) | | | | 0.133 *** (0.0462) | 0.176 *** (0.0489) |
| Prudential regulation of banks | | | 0.0462 ** (0.0193) | | | 0.0620 *** (0.0200) | 0.0509 ** (0.0210) |
| Removal of banks entry barriers | | | | −0.0695 *** (0.0186) | | −0.0918 *** (0.0250) | −0.0828 *** (0.0278) |
| Loosening of credit controls *(lagged)* | | | | | 0.0352 ** (0.0152) | 0.0451 (0.0336) | |
| Removal of credit ceilings | | | | | | | 1.034 * (0.617) |
| Constant | −1.648 *** (0.216) | −1.720 *** (0.201) | −1.304 *** (0.159) | −0.815 *** (0.113) | −1.142 *** (0.146) | −1.760 *** (0.210) | −2.691 *** (0.669) |
| Observations | 331 | 331 | 331 | 323 | 323 | 331 | 275 |
| Number of countries | 18 | 18 | 18 | 18 | 18 | 18 | 15 |

Standard errors in parentheses; *** $p < 0.01$, ** $p < 0.05$, * $p < 0.1$; reform indicators are extended up to 2018 using the methodology described in the text.

**Table 6.** Robustness check: financial sector reforms and unemployment outflows

| | Dependent Variable: Unemployment Outflows | | | | | | | | |
|---|---|---|---|---|---|---|---|---|---|
| | **(6.1)** | **(6.2)** | **(6.3)** | **(6.4)** | **(6.5)** | **(6.6)** | **(6.7)** | **(6.8)** | **(6.9)** |
| Outflows *(lagged)* | 0.446 *** (0.0871) | 0.437 *** (0.0887) | 0.426 *** (0.0899) | 0.442 *** (0.0865) | 0.419 *** (0.0851) | 0.426 *** (0.0863) | 0.410 *** (0.0741) | 0.676 *** (0.0350) | 0.668 *** (0.0367) |
| Output gap | 0.0270 *** (0.00714) | 0.0268 *** (0.00714) | 0.0241 *** (0.00723) | 0.0277 *** (0.00704) | 0.0288 *** (0.00696) | 0.0279 *** (0.00701) | 0.0301 *** (0.00657) | 0.0411 *** (0.00522) | 0.0397 *** (0.00522) |
| Real long term interest rates | −0.0575 *** (0.0100) | −0.0592 *** (0.0103) | −0.0612 *** (0.0104) | -0.0565 *** (0.00969) | −0.0548 *** (0.00926) | −0.0562 *** (0.00951) | −0.0530 *** (0.00778) | −0.0258 *** (0.00597) | −0.0278 *** (0.00609) |
| Real wage growth *(lagged)* | −0.0177 *** (0.00357) | −0.0183 *** (0.00355) | −0.0164 *** (0.00356) | -0.0180 *** (0.00358) | −0.0173 *** (0.00352) | −0.0166 *** (0.00360) | −0.0143 *** (0.00354) | −0.0177 *** (0.00360) | −0.0167 *** (0.00360) |
| Real share price variation | 0.0872** (0.0370) | 0.0858** (0.0370) | 0.0886 ** (0.0368) | 0.0847** (0.0372) | 0.0732 ** (0.0370) | 0.0774 ** (0.0372) | 0.0468 (0.0344) | 0.114 *** (0.0326) | 0.115 *** (0.0326) |
| Financial reforms | 0.0147 *** (0.00523) | | | | | | | | |
| Loosening of credit controls | | 0.0544 *** (0.0169) | | | | | | | |
| Privatization of banks | | | 0.109 *** (0.0211) | | | | | | 0.0479 *** (0.0127) |
| Prudential regulation of banks | | | | | 0.0198** (0.0101) | | | | |
| Security markets deregulation | | | | | −0.233 *** (0.0632) | | −0.281 *** (0.0648) | 0.108 * (0.0611) | 0.104 * (0.0610) |
| Capital account opening *(lagged)* | | | | | | 0.126 *** (0.0419) | 0.117 *** (0.0396) | 0.121 *** (0.0417) | 0.130 *** (0.0424) |
| Prudential regulation of banks *(lagged)* | | | | | | | 0.0202 ** (0.0103) | 0.0476 *** (0.0116) | 0.0329 *** (0.0117) |
| Removal of bank entry barriers | | | | | | | | −0.230 *** (0.0272) | −0.232 *** (0.0276) |
| Constant | −1.181 *** (0.227) | −1.053 *** (0.188) | −1.180 *** (0.201) | −0.955 *** (0.159) | -0.262 (0.165) | −1.315 *** (0.250) | −0.542 ** (0.216) | −0.640 *** (0.225) | −0.736 *** (0.233) |
| Observations | 423 | 423 | 423 | 423 | 423 | 423 | 423 | 423 | 423 |
| Number of countries | 22 | 22 | 22 | 22 | 22 | 22 | 22 | 22 | 22 |

Standard errors in parentheses; *** $p < 0.01$, ** $p < 0.05$, * $p < 0.1$; reform indicators are extended up to 2018 using the methodology described in the text.

## 6. Reform Scenarios and Unemployment Dynamics

Most advanced economies have embarked on reforms to tighten the regulation of financial markets in the aftermath of the GFC. The estimates presented in the previous section suggest that such changes to the regulatory environment in which banks and financial investors operate is likely to have statistically-significant effects on unemployment dynamics. However, not all reforms will also have an economically-relevant impact, nor will the impact of different reforms necessarily affect job dynamics in a similar fashion. For instance, the estimations show that both capital account opening and improved banking supervision have unambiguously positive effects on employment, lowering job destruction and increasing job creation. On the other hand, deregulation of the securities markets does mainly seem to increase labor market turbulence, increasing both unemployment in- and out-flows. Finally, deepening derivatives markets will hamper outflows, but leave inflows (statistically) unaffected, thereby increasing unemployment stocks and duration. Taken together, the impact of financial sector reforms will, therefore, depend on the concrete packages that countries are adopting and the relative weight they put on individual reforms in these three areas.

Whatever the benefits of proper financial sector regulation, policy makers are not free in choosing the optimal level of regulation that would correspond to theory. Besides the fact that a single, optimal reform package may actually not exist given the different layers of financial regulation, at least three considerations will limit the capacity of policy makers to reform financial markets:

- The financial and economic recovery actually complicates the task of substantial regulatory reform of financial markets. Political pressure for reforms wears off as business activities resume. The immediate sense of urgency recedes, making policy makers more lenient when putting forward an encompassing reform agenda. In addition, even though the crisis had somewhat reduced the political influence of financial firms in the immediate aftermath, as soon as the outlook improved, financial sector lobby groups started to gain a stronger political voice again. Finally, financial sector (re-)regulation will take place in a substantially different macroeconomic environment. Even 10 years after the crisis, the risk appetite of investors has resumed only partially. Over the longer term, however, investors are likely to re-evaluate their environment and consider investing in higher yielding and more risky assets.

- At the same time, countries and policy makers are limited in their action by the high level of public debt that has accumulated during the crisis and has not been brought down since. This will reduce their scope for action and hence the extent to which they can effectively introduce any kind of regulation without regard to the interests of capital owners and their own financiers. In the past, periods of rapid increase in public sector debt have often preceded periods of financial deregulation. In other words, even if it were possible to identify ex ante the optimal package of financial sector regulation, such a reform bundle is unlikely to be implemented ex post as policy makers rely heavily on financial markets to (re)finance their high and still increasing debt levels.

- Finally, regulatory competition between jurisdictions prevents countries from implementing all measures deemed necessary for fear of losing (financial sector) market share to competitors. As countries compete to attract financial firms through favorable regulatory conditions, overly stiff prudential regulation may hamper further growth of the financial sector. Highly-qualified staff may consider moving to different locations with a more attractive tax and regulatory environment (for instance regarding bonus regulation). Similarly, financial firms may consider moving their activities to jurisdictions where limitations on leverage and credit growth are less stringent, offering their services to clients abroad or arbitraging across different regulatory conditions through branching.

Considering different areas of financial regulation and the existence of political barriers to financial reform, four different reform scenarios are being considered in the following in order to evaluate their impact on employment dynamics. Such reform packages—in contrast to individual reforms—are likely to have larger, macroeconomic effects: First, changes to prudential regulation will have implications

on financial market stress and volatility. Related, financial market regulation influences the cost of capital through the risk-free rate, as well as the development of stock market valuations. As regards reforms to international capital flows, changes in the international financial architecture may impact both international capital and trade flows. By constructing alternative reform scenarios, this last section offers a contribution to a global assessment of real economy effects of different reform packages. The following table summarizes the different assumptions that are underlying the impact analysis of the reform scenarios on labor markets.

In order to illustrate the implications of different reform scenarios for labor market outcomes, we compare three different reform scenarios with the baseline scenario of unreformed financial markets (see Table 7). The reform options chiefly focus on enhanced prudential regulation of the domestic banking sector and the regulation of international capital flows. As argued elsewhere (see International Institute for Labour Studies 2010, Chp. 5), these two areas are likely to be reformed independently of each other, with no guarantee that policy reforms will be coordinated between national and international regulators. The impact of the reforms will be felt along different dimensions (see Table 3, equation (3.11), and Table 4, equation (4.9)): All reform scenarios come with reduced financial stress in comparison to the baseline scenario of an unreformed financial sector. When reforms are concentrated on the domestic market, the impact will be predominantly felt in a less vibrant stock market with lower share price growth and higher capital costs. When reforms are concentrated on the international market, both international capital flows and world trade will grow less rapidly.

On the basis of these assumptions regarding the implications of the four scenarios for financial sector development, an estimation has been carried out as to the likely impact on employment dynamics in a typical advanced G20 country (see Table 8). Both equations replicate the results of the previous sections in addition to a fuller specification including the IMF Financial Stress Index to indicate exogenously-driven shocks to the financial system.

Together, Tables 7 and 8 allow quantifying the likely impact of the reform options on the main determinants that influence unemployment in- and out-flows as described in the previous section. Figure 5 assumes that—starting in 2010—the real value of outstanding shares would increase permanently by 10%, that annual trade growth would continue at 10%, and that capital flows would increase by 10% per year. No further securities or prudential regulation in the banking sector would be introduced. At the same time, this scenario assumes a unit increase in global financial stress as measured by the indicator produced by Balakrishnan et al. (2009). The quantitative scenario assumes not only an impact of financial market stress on employment creation, but also on labor supply (through a discouraged worker effect). In particular, according to the underlying estimates, labor force growth is permanently depressed by one percentage point if the financial market stress indicator raises by a unit increase. As the following figure shows, despite this additional financial market stress, employment growth continues to recover, thanks to strong trade and share prices growth. After a peak in 2015, it will gradually return to its long-run trend rate at around 1.7%, in line with labor force growth in this region (no change in demographics have been assumed in these simulations).

**Table 7.** Exit scenarios from the crisis: assumptions about macroeconomic implications.

| International Capital Flows / Domestic Financial Markets | UNREFORMED | TIGHTENED REGULATION |
|---|---|---|
| **UNREFORMED** | *Scenario I:*<br>Permanent increase in financial stress<br>Return to highly-valued shares<br>Continued export and import growth<br>High international capital flows | *Scenario III:*<br>Moderate reduction in financial stress<br>Moderate reduction or stable share prices<br>Further export and import growth<br>Moderate increase in international capital flows |
| **TIGHTENED REGULATION** | *Scenario II:*<br>Moderate increase in financial stress<br>Stable share prices<br>Slower trade growth<br>Reduced international capital flows | *Scenario IV:*<br>Permanent reduction in financial stress<br>Lower real share prices<br>Slower growth of world trade<br>Reduction in international capital flows |

**Table 8.** Scenario estimations.

| | Outflows (lagged) | Output gap | Real long term interest rate | Wage-interest rate (yearly change) | Real share price (yearly change) | Investment growth (per cent) | Capital openness | Securities markets | Prudential regulation | Financial stress index (change) |
|---|---|---|---|---|---|---|---|---|---|---|
| **Outflows** | 0.556 *** (0.115) | 0.019 * (0.010) | −0.030 ** (0.015) | −0.105 * (0.055) | 0.031 (0.060) | 12.934 *** (3.60) | 0.205 *** (0.059) | −0.116 * (0.064) | 0.053 *** (0.017) | −0.026 *** (0.008) |
| | Inflows (lagged) | Output gap | Labor force growth (p.a.) | Total factor productivity growth (lagged) | | Banking sector entry barriers | Capital openness | Securities markets | Prudential regulation | Financial stress index (lagged) |
| **Inflows** | 0.867 *** (0.023) | −0.051 *** (0.010) | 1.600 ** (0.672) | 0.175 * (0.103) | | −0.059 ** (0.025) | $-2.3 \times 10^{-4}$*** ($6.2 \times 10^{-5}$) | 0.048 ** (0.023) | 0.028 *** (0.010) | 0.010 *** (0.004) |

Note: The table displays the estimations for the unemployment outflow and inflow equation underlying the scenario analysis. Estimations are based on GMM estimations. Both equations contain constants (not displayed) and pass the Sargan test for over-identification of the instruments.

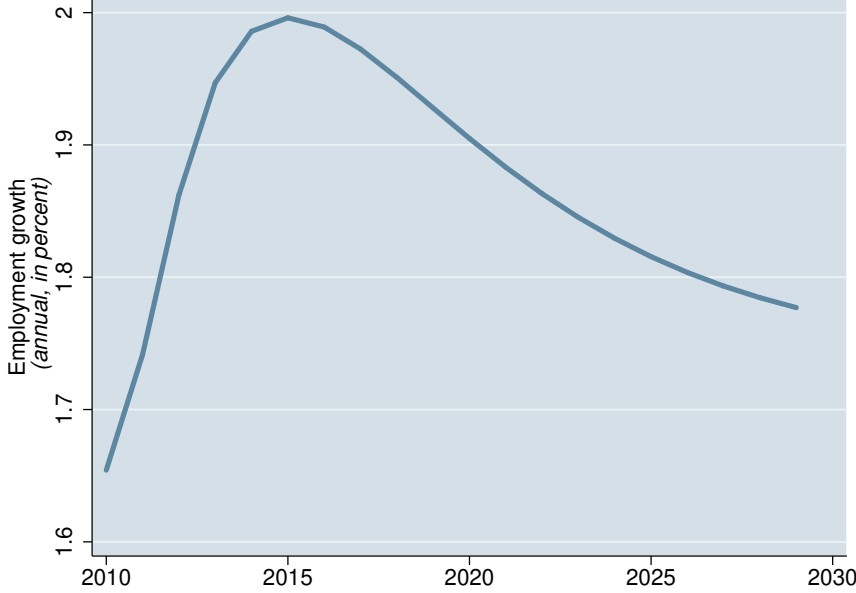

**Figure 5.** Employment dynamics with unreformed financial markets. Note: The chart presents the evolution of employment growth in the baseline scenario of unreformed financial markets.

In comparison, the three other scenarios assume—each to a different degree—a further tightening of either securities or banking sector regulation, whereby Scenario IV makes the strictest assumptions about the evolution of these indicators (see Figure 6). Trade is expected to decline in Scenarios II and IV, whereas financial market stress (and the real value of outstanding shares) declines only in Scenario III and IV, thanks to the introduction of tighter domestic regulation. As the following chart demonstrates in the long-run, the effects on employment are negative in the short-run, as expected, although certainly much less than what has been predicted by others elsewhere (Institute of International Finance 2010). Already after three years, some improvements can be felt, in particular due to the decrease in financial sector volatility. Under Scenario II, where this effect is weakest, the adverse effects from reduced dynamics in world trade and financial market activity will keep the employment growth rate permanently below the baseline rate of unreformed financial markets. However, when policy makers show more ambition, in particular as regards domestic re-regulation and the supervisory framework of the banking sector, stronger positive effects for employment creation can be expected. In other words, the increase in costs resulting from stricter banking sector regulation can be considered moderate in comparison to the benefits from lower financial market volatility, a point also made by Admati et al. (2013) or Kashyap et al. (2010). Taken together, these results suggest that financial sector regulation, had it been in place already at the time of the crisis, would not only have helped stabilize the economy, but would have also supported a faster recover of employment growth.

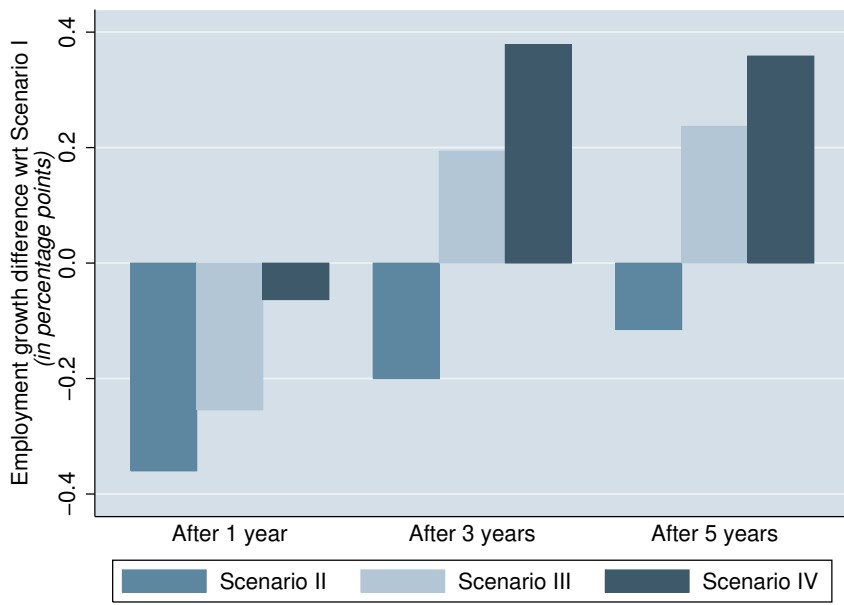

**Figure 6.** Comparisons of employment dynamics: Scenarios II–IV vs. Scenario I. Note: The chart compares employment growth rates of different reform scenarios with the baseline. The bars represent differences in annual employment growth rates in percentage points.

## 7. Conclusions

This article is the first to present a detailed analysis of the effects of financial market development and reforms on labor market dynamics. Against the backdrop of renewed discussions on the benefits of (strict) financial market regulation, it discusses the likely impact on labor market dynamics of reforms in credit provision, interest rate controls, capital account openness, and prudential regulation. As the article demonstrates, such financial market reforms could have brought about substantial benefits for job creation had these measures been implemented in a coherent manner prior to the crisis.

The article starts by presenting a theoretical framework through which to analyze the empirical implications of financial market development and regulation. On the basis of the ILO's database on unemployment flows, we then present the impact of financial market variables on unemployment outflows and inflows, demonstrating the significant effects that well-developed, well-regulated financial markets have on both margins of labor market adjustment.

When analyzing more encompassing financial sector reforms, the article shows that some negative effects might have been expected from tighter regulation in the short-run. However, over the medium-run, employment creation would have strongly benefited from the reduced volatility that a more elaborate framework for securities, banking supervision, and capital controls would have brought. In this regard, policy-makers should not lose the momentum for pushing through with the implementation of already agreed-upon reforms such as those to strengthen capital adequacy rules by the Basel III framework, in order to reduce disproportionate leverage and excessive incentives for risk-taking within the banking sector. As indicated by our extended financial reform index, there is some worry that the reform effort displayed in the aftermath of the global financial crisis is already losing momentum. In this regard, continuing the implementation of these regulatory reforms would greatly reduce the still very high levels of uncertainty among market participants, and also reduce volatility and risk premia, thereby supporting output and employment growth. The benefits of financial sector reform for the real economy will be greatest when they are implemented in a coordinated fashion, reforming both domestic financial markets and the international financial system. This article shows that such reforms are feasible and likely to yield the expected positive results for employment creation.

**Funding:** This research received no external funding.

**Acknowledgments:** The author gratefully acknowledges helpful comments from Inessa Love (University of Hawaii) on an earlier version, as well as discussions with Matthieu Charpe and Uma Rani (both ILO). Daphne Halikiopoulou (University of Reading) provided excellent editorial support. Federico Curci helped in preparing the data and giving overall research assistance. The views expressed in this paper are those of the author and cannot be imputed to the International Labour Organization.

**Conflicts of Interest:** The author declares no conflict of interest.

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
