# Peer review of "Finance and Jobs: How Financial Markets and Prudential Regulation Shape Unemployment Dynamics"

_jrfm, doi:10.3390/jrfm12010020_

Reviewer 1 Report

This paper deals with an important and provides interesting evidence. The paper is well constructed. This paper is generally interesting and well-implemented; I do, however, note the following:

1. The introduction should be extended and the novelty of the contribution better stressed.

2. The literature review section must be improved with papers that highlight the topic. The literature review section should end with summary and synthesis.

3. The reference citation and organization of the paper doesn't meet the requirements of the journal.

4. The analysis of the calculations could be supported with deeper argumentation.

5. The conclusions could be supported with results of previous research. 

Author Response

Dear Referee,
Many thanks for your time and valuable comments! We have revised our manuscript according to the review reports step by step. Please check it.
Thanks again!
Best regards

(1) The introduction has been revised and the novelty of the approach be highlighted.

(2) The literature review has substantially be revised and extended.

(3) Wherever possible, the bibliography style has been brought in line with the journal. For the finalization of the paper, a BST-file would be helpful that might correct remaining mistakes. 

(4) The analysis of the results has been better motivated through the extension of the literature review. 

(5) The conclusion has been revised and reference to previous research be included.

Reviewer 2 Report

Summary: This paper investigates the relationship between financial market development and unemployment flows. The data is based on OECD country level data for the period 1970-2009. Dynamic panel data estimations show that financial sector reforms are significantly associated with unemployment in- and outflows. I like the general approach of the authors. However, the database is outdated and the first-difference GMM estimator is not appropriate when the number of time periods is relatively large as compared to the number of cross-sectional units. Also, the first-difference GMM estimator poorly performs when explanatory variables are almost time invariant. In addition, the data set is quite old, and references must be up to date. Below there are number of comments that must be addressed.

General comments

The contribution must be clearly stated in the introduction. Write something like this:

The main contribution is that this is the first analysis to examine the impact of financial sector reform after the financial and economic crisis using panel data for developed countries.

The theoretical model in Section 3 is not new. We do not need a replication of the Wasmer and Weil model. Focus on the new aspects of the paper that represents the empirical part. I suggest condensing this material.

The first difference Arellano Bond GMM estimator is a large sample estimator (reference is missing in the reference list). It is appropriate when the number of cross-sectional units is large, and the number of time periods is short. Here N is<20 3="" and="" t="" in="" table="">N and N==8(!).The AB estimator cannot be used when variables are almost time invariant. I can see that there is limited time variation in the right-hand variables. The system GMM estimator by Blundell and Bond (1998) performs much better when variables are almost time invariant. However, you need a large sample size (min 50). In general, the pooled mean group estimator developed by Pesaran et al. (1999) performs better when T>N. However, variables must be time varying. You might also consider dynamic ML (Hsiao et al., 2002) and spilt the sample period into three subsamples.

The database is quite old and must be updated. More recent data on financial sector reforms should be available. There is a change in the financial sector variables after 2009. It would be interesting to know the difference in the estimations.

Also, you motivate the paper with a statement of the economic and financial crisis but the data ends in 2009. Rewrite the introduction. Also update the reference list. There is only one reference after 2010.

The Plümper and Troeger (2007) Fixed Effects Vector Decomposition (FEVD) approach is not an alternative when variables are almost time invariant.

See here Greene (2011): “The FEVD estimator proposed by Plümper and Troeger (2007) is illusory.  The development of the estimator exploits an interesting algebraic result that reaches an old conclusion via a new path – the estimator is the original least squares dummy variable estimator. The claimed efficiency gains under their assumptions are produced by using an erroneous result”

Literature

Arellano, M., & Bond, S. (1991). Some tests of specification for panel data: Monte Carlo evidence and an application to employment equations. The review of economic studies, 58(2), 277-297.

Blundell, R., & Bond, S. (1998). Initial conditions and moment restrictions in dynamic panel data models. Journal of econometrics, 87(1), 115-143.

Greene, W. (2011), ‘Fixed effects vector decomposition: a magical solution to the problem of time invariant variables in fixed effects models?’, Political Analysis, Vol 19, No 2, pp 135–146.

Hsiao, C., Pesaran, M. H., & Tahmiscioglu, A. K. (2002). Maximum likelihood estimation of fixed effects dynamic panel data models covering short time periods. Journal of Econometrics, 109(1), 107-150.

Pesaran, M. H., Shin, Y., & Smith, R. P. (1999). Pooled mean group estimation of dynamic heterogeneous panels. Journal of the American Statistical Association, 94(446), 621-634.

Plümper, T., & Troeger, V. E. (2007). Efficient estimation of time-invariant and rarely changing variables in finite sample panel analyses with unit fixed effects. Political Analysis, 15(2), 124-139.

Author Response

Dear Referee,
Many thanks for your time and valuable comments! We have revised our manuscript according to the review reports step by step. Please check it.
Thanks again!
Best regards

(1) The introduction and literature review have been substantially reviewed and extended to highlight the novelty of the approach. In particular, the relevance of looking at labour market implications of reforms have been stressed more.

(2) The presentation of the theoretical model is already a condensed version of the Wasmer and Weil approach. The discussion has been focused on the development of empirically testable equations relevant for the main argument of the article.

(3) The use of the Arellano-Bond system GMM estimator in the particular context has been better motivated and tied to previous approaches in the literature. The specific issue of proliferation of instruments that can arise in this approach has been highlighted and the solution to this issue indicated.

(4) The reason for using a database (for financial sector reforms) that stops in 2005 has been explained and the motivation for the results highlighted accordingly. In particular, the results should be interpreted as indicating the potential effects of financial sector regulation had it been in place prior to the global financial crisis. 

(5) The use of the estimator suggested by Pluemper and Troeger has been dropped and results in section 6 re-estimated using system GMM. Results of the simulations remain essentially unchanged. 

Round  2

Reviewer 2 Report

Referee report

The authors have considered many comments and improved the paper. However, the authors did not address the most important critical point regarding the estimation method. This is a great weakness of the work. Also, the authors confuse the first-difference Arrelano Bond estimator with the system GMM estimator developed by Blundell and Bond (1998). The authors use a small macro panel with T=19 or 20 and N=20. In some estimates N is equal to 5. Neither the first difference Arrelano-Bond nor the system GMM are suitable for this kind of panel data. Bond himself in a recent contribution (Bond and Xing, 2015) uses the pooled mean group estimator developed by Pesaran et al. (1999). This estimation method is suitable when T is relatively long and N is small. The pooled mean group estimator also has the advantage that you can test whether the coefficients are the same across countries. Authors have to re-estimate the equations using the pooled mean group estimator and remove the first difference GMM estimation results.

By the way, nobody uses the first difference GMM estimation method anymore because lagged values are poorly correlated with its first differences if the variables are very persistent.

References

Bond, S., & Xing, J. (2015). Corporate taxation and capital accumulation: Evidence from sectoral panel data for 14 OECD countries. Journal of Public Economics, 130, 15-31.

Pesaran, M. H., Shin, Y., & Smith, R. P. (1999). Pooled mean group estimation of dynamic heterogeneous panels. Journal of the American Statistical Association, 94(446), 621-634.

Author Response

I am aware of the problems that the first difference GMM estimator poses, which is why the system GMM estimator has been preferred throughout the estimations and I have ensured that the Sargan overidentification test has been validated.

Unfortunately, due to the large number of co-variates more sophisticated techniques such as the one suggested are not available. The only alternative would be to revert to standard fixed effect OLS with lagged dependent variables, which have not been the preferred approach.

I take comfort in the fact that the baseline specification is stable even when extending the sample with a larger number of countries and years as in the suggested robustness check section 5.3 where the financial reform indicator has been extended.